# Bigram Subnetworks: Mapping to Next Tokens in Transformer Language Models

**Tyler A. Chang**
Department of Cognitive Science
University of California San Diego
`tachang@ucsd.edu`

**Benjamin K. Bergen**
Department of Cognitive Science
University of California San Diego
`bkbergen@ucsd.edu`

## Abstract

In Transformer language models, activation vectors transform from current token embeddings to next token predictions as they pass through the model. To isolate a minimal form of this transformation, we identify language model subnetworks that make bigram predictions, naive next token predictions based only on the current token. We find that bigram subnetworks can be found in fully trained language models up to 1B parameters, and these subnetworks are critical for model performance even when they consist of less than 0.2% of model parameters. Bigram subnetworks are concentrated in the first Transformer MLP layer, and they overlap significantly with subnetworks trained to optimally prune a given model. Mechanistically, the bigram subnetworks often recreate a pattern from the full models where the first layer induces a sharp change that aligns activations with next token predictions rather than current token representations. Our results demonstrate that bigram subnetworks comprise a minimal subset of parameters that are both necessary and sufficient for basic next token predictions in language models, and they help drive the transformation from current to next token activations in the residual stream. These subnetworks can lay a foundation for studying more complex language model circuits by building up from a minimal circuit.[1]

## 1 Introduction

Modern Transformer language models predict next tokens from text. However, developing reliable and interpretable explanations for how they do this is an open area of research. In particular, *mechanistic interpretability* research aims to identify features in model activations and operations in model parameters that explain how language models process text to make predictions. Ideally, it would be possible to decompose language model behavior into component circuits, each of which performs some interpretable operation in the model that can be isolated from other operations. Indeed, an extensive body of research has sought to identify interpretable circuits in language models, including induction heads, name mover heads, copy suppression heads, indirect object identification (IOI) circuits, and other types of reasoning circuits (Olsson et al., 2022; Wang et al., 2023; Geva et al., 2023; Lepori et al., 2023b; McDougall et al., 2024; Merullo et al., 2024a,b; Lindsey et al., 2025).

Circuit behaviors are often verified through ablations or interventions, where a target behavior should change when the circuit is ablated or modified in the full model (e.g. Finlayson et al., 2021; Meng et al., 2022; Stolfo et al., 2023; McDougall et al., 2024; Tigges et al., 2024; Zhang et al., 2024; Todd et al., 2024; Sankaranarayanan et al., 2024; AlKhamissi et al., 2025). This evaluates the *necessity* of the circuit for the target behavior, but not its *sufficiency*: the circuit should also induce the behavior when added to a minimal circuit whose behavior is already understood. Both necessity and sufficiency

---

[1] Code and trained bigram subnetworks at: `https://github.com/tylerachang/bigram-subnetworks`.

39th Conference on Neural Information Processing Systems (NeurIPS 2025).

are critical to delineating a circuit's functional role. For example, a circuit that uniquely performs a *sub*operation of the target behavior is necessary but not sufficient (Merullo et al., 2024a); a circuit that redundantly performs the target behavior is sufficient but not necessary (Wang et al., 2023).[2] Observing or ablating circuits in a full model does not permit evaluation of circuit sufficiency, because the circuit might rely on features or suboperations performed by other parts of the model. To establish both necessity and sufficiency, we must also observe the circuit's behavior when operating only over some well-understood minimal circuit. Previous work has used an empty circuit as the minimal circuit (e.g. Wang et al., 2023; Conmy et al., 2023; Hanna et al., 2024; Nikankin et al., 2025; Marks et al., 2025; Mueller et al., 2025), but this does not address how different circuits *build upon* one another (Lepori et al., 2023b; Merullo et al., 2024a; Miller et al., 2024).

So where should we start when building language models up from minimal circuits? To start, we need a simple and clearly interpretable behavior, but one that is well-defined over the entire input space (i.e. a behavior that has a predicted output for every possible input). *Bigram* predictions $P(w_i|w_{i-1})$, next token predictions conditioned only on the current token, are such a behavior. They are also arguably the simplest nontrivial next token predictions, and there is evidence that Transformer language models overfit to bigram predictions early in pretraining (Chang & Bergen, 2022; Choshen et al., 2022; Chang et al., 2024). Isolating how Transformer language models make bigram predictions could help researchers understand how language models make next token predictions at a basic and interpretable level, and bigram subnetworks in larger models could serve as a basis upon which to study the effects of more complex circuits in future work.

Thus, in this paper, we find subnetworks that recreate bigram predictions in Transformer language models up to 1B parameters, and we find that they:

1. Consist of roughly 10M parameters regardless of model size, consistently reaching bigram surprisal correlations of $r > 0.95$ (§3).
2. Are concentrated in the first MLP layer throughout pretraining (§4).
3. Recreate key properties of the residual stream, such as a rotation from current to next token activations in the first Transformer layer (§5).
4. Drastically hurt language modeling performance when ablated, and overlap significantly with subnetworks trained to optimally prune a given model (§6).

Our results suggest that bigram subnetworks comprise a minimal subset of parameters that serve as the core of a Transformer language model. Concretely, despite being highly sparse, they are both *necessary* and *sufficient* for basic next token predictions in language models.

## 2   Background and Related Work

There is good reason to expect Transformer language models to exhibit bigram subnetworks, based on previous research drawing connections between Transformer language models and traditional $n$-gram models. An $n$-gram model predicts each next token $w_i$ based only on the previous $n-1$ tokens: $P(w_i|w_{i-1}, ...w_{i-n+1})$. Transformer models trained on formal languages or directly on $n$-grams can learn $n$-gram distributions (Elhage et al., 2021; Svete & Cotterell, 2024; Svete et al., 2024), and they can recognize $n$-grams in context using specialized attention heads (Akyürek et al., 2024). When trained on natural language, Transformer models overfit to $n$-gram probabilities for increasing $n$ early in pretraining (Chang & Bergen, 2022; Choshen et al., 2022; Chang et al., 2024), and their predictions can be approximated to some degree using $n$-gram rule sets (Nguyen, 2024). Smoothed $n$-grams alone achieve fairly high next token prediction accuracy for large $n$ (Liu et al., 2024).

To investigate how Transformer language models trained on the standard language modeling objective make bigram predictions mechanistically, we find bigram subnetworks using *continuous sparsification* (§3.1; Savarese et al., 2020; Lepori et al., 2023b,a). Lepori et al. (2023b) apply this method to find subject-verb agreement and reflexive anaphor agreement subnetworks in BERT-small; we extend this approach to larger autoregressive models and the more general function of bigram prediction. In analyzing the subnetworks we find, we draw on work studying the *residual stream* in Transformer

---

[2]We define circuit necessity and sufficiency in more detail in §A.1. There, we also compare to the complementary notions of circuit *faithfulness*, *completeness*, and *minimality* from Wang et al. (2023). Notions of circuit necessity and sufficiency are also closely related to *task demands* in language models, where certain capabilities are necessary but not sufficient for some tasks (Hu & Frank, 2024).

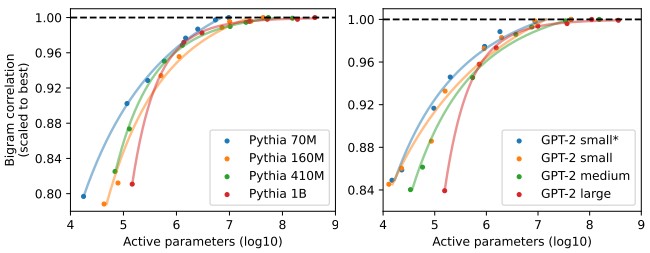

| Model | Bigram $r$ | Full model |
|---|---|---|
| Pythia 70M | **0.961** | 0.737 |
| Pythia 160M | **0.964** | 0.690 |
| Pythia 410M | **0.983** | 0.650 |
| Pythia 1B | **0.987** | 0.632 |
| GPT-2 small* | **0.987** | 0.680 |
| GPT-2 small | **0.979** | 0.624 |
| GPT-2 medium | **0.985** | 0.582 |
| GPT-2 large | **0.986** | 0.583 |

Figure 1: Left, center: bigram surprisal correlations for subnetworks with different numbers of active parameters (excluding embedding parameters), for different models. Bigram correlations are scaled to the highest bigram correlation for any subnetwork trained for that model. Correlations plateau at roughly 10M active parameters regardless of model size. Right: bigram surprisal correlation $r$ for the highest-correlation subnetwork vs. the full model for each model. GPT-2 small* indicates the GPT-2 small replication from Chang et al. (2024).

language models. The residual stream hypothesis theorizes that Transformer models read from and write to an activation space that remains relatively stable across layers (Nostalgebraist, 2020; Geva et al., 2021, 2022; Dar et al., 2023; Belrose et al., 2023). This shared activation space is often thought to align roughly with next token embedding space after the first layer, an observation we quantify in §5. Interestingly, this transformation to next token space seems to occur much earlier than suggested by Voita et al. (2024), who find neurons across all layers that activate for specific tokens to promote corresponding next token (bigram) predictions. In our work, we precisely identify language model parameters that are necessary and sufficient for bigram predictions, and we highlight their importance to a language model's performance.

## 3   Finding Bigram Subnetworks

We define a bigram subnetwork as a subset of language model parameters such that if all other parameters are set to zero, then the model mimics the bigram prediction $P(w_i|w_{i-1})$.[3] In other words, bigram subnetworks are *sufficient* for bigram predictions. In this section, we find that such bigram subnetworks exist even in language models over 1B parameters, and they consistently plateau in bigram reconstruction ability at roughly 10M active parameters (§3.2). In future sections, we verify the validity and consistency of these bigram subnetworks through structural analyses across checkpoints (§4), mechanistic analyses (§5), and subnetwork ablations (§6).

### 3.1   Method: Continuous Sparsification

We focus on Pythia (Biderman et al., 2023) and GPT-2 models (Radford et al., 2019), with sizes up to Pythia 1B and GPT-2 large (1.01B and 774M parameters respectively). For analyses across check-points, we focus on Pythia 160M, Pythia 1B, and a re-trained GPT-2 small model with checkpoints from Chang et al. (2024). All bigram subnetworks are trained and evaluated using English web text data from OSCAR (Abadji et al., 2021).

To find bigram subnetworks, we use continuous sparsification (Savarese et al., 2020; Lepori et al., 2023b), which optimizes a mask $M$ over frozen model parameters to minimize a given loss function. Continuous sparsification optimizes one real-valued parameter $m \in (-\infty, +\infty)$ per original model parameter; each $m$ is mapped into the interval $(0, 1)$ using a sigmoid function, to create entries in the mask $M$. The temperature of the sigmoid function is decreased throughout subnetwork training such that mask values approach 0 and 1, eventually resulting in a mask $M$ that can be converted to a binary mask. Then, $M$ defines a subset of model parameters that approximately minimizes the target loss function. To optimize $M$ to mimic the bigram distribution $P$, we use the following loss function for model input $x$:

$$\text{Loss}(M, x) = \text{CrossEntropy}\Big(P(x), \text{MaskedModel}_M(x)\Big) + \lambda \frac{||M||_1}{|M|} \tag{1}$$

---

[3]We call this a "subnetwork" rather than a "circuit" to emphasize that bigram subnetworks are not necessarily localized to specific attention heads and layers that execute intermediate suboperations.

As in Savarese et al. (2020), $\lambda$ weights the L1 penalty term $||M||_1$, which enforces sparsity in the mask $M$. We normalize by the total number of trainable mask parameters $|M|$, and we train subnetworks for $\lambda \in [0, 1, 5, 10, 50, 100, 500, 1000]$ to evaluate the effects of sparsity on subnetwork performance. We learn the mask $M$ over all model parameters except input and output embedding parameters and layernorm parameters. We train each subnetwork on sequences of 128 tokens with batch size 32 and learning rate 5e-5, and we set the mask sigmoid temperature to divide by 1.001 per training step (dictating how fast $M$ approaches a binary mask). We find that these hyperparameters generally allow subnetwork training to converge for all models tested, and the resulting subnetworks have similar evaluation loss to subnetworks trained under different hyperparameter settings. Subnetwork training and convergence details are in §A.2.

### 3.2 Existence of Bigram Subnetworks

As shown in Figure 1, for all models, continuous sparsification is able to find subnetworks whose surprisals (i.e. token-level losses or negative log-probabilities) correlate highly with those of bigrams (Pearson's $r > 0.95$). We measure surprisal correlations rather than cross-entropies themselves because they are more interpretable across models, they are more efficient to compute when bigram surprisals can be cached, and we find that they produce similar patterns of results. Notably, we find that bigram correlations plateau at roughly 10M active non-embedding parameters regardless of model size (Figure 1, left, center).[4] This suggests that bigram subnetwork "size" is roughly independent of full model size, and thus the bigram subnetwork comprises a much smaller proportion of model parameters in larger models. For example, in Pythia 1B, a subnetwork containing only 0.17% of non-embedding parameters reaches a bigram surprisal correlation of $r = 0.959$.

## 4 Persistence and Structure Throughout Pretraining

We now study the distribution and structure of the active parameters found in bigram subnetworks. Because language models have been found to overfit to bigram predictions early in pretraining (Chang & Bergen, 2022; Choshen et al., 2022; Chang et al., 2024), we particularly consider whether this structure changes throughout pretraining. We find that bigram subnetworks persist during pretraining long after the full model has diverged from bigram predictions (§4.1), and they consist primarily of MLP parameters in the first Transformer layer (§4.2).

### 4.1 Bigram Subnetworks Persist but Decompress Throughout Pretraining

We start by investigating whether the number of parameters required for bigram subnetworks changes throughout pretraining. To do this, we train bigram subnetworks as in §3.1 with sparsity term $\lambda \in [0, 1, 10, 100, 500]$ for 16 checkpoints in Pythia 160M and 1B, and for 21 GPT-2 small checkpoints from Chang et al. (2024). We fit a power law approximating bigram surprisal correlation from number of active parameters (as in Figure 1) for each checkpoint; we use this curve to estimate the bigram correlation that can be achieved using 500K, 1M, or 10M active parameters at each checkpoint.

Results for Pythia 1B are in Figure 2 (left), with qualitatively similar results for other models in §A.6. We find that bigram subnetworks exist long after the models begin to diverge from bigram predictions, demonstrated by high bigram correlations even after the full model exhibits a drop in bigram surprisal correlation. Notably, bigram subnetworks peak in bigram correlation over two thousand steps *after* the full model overfits to bigram predictions, suggesting that language models continue to learn bigram distributions even after that behavior becomes less observable from the full model.

However, once the bigram subnetworks peak in their ability to recreate the bigram distribution (roughly step 4K in Figure 2, left), they start to take more parameters to reach such high bigram correlations. For example, 500K active parameters can reach a bigram surprisal correlation of $r = 0.949$ at checkpoint 4K but only $r = 0.919$ at checkpoint 143K (Figure 2, left). This suggests that while bigram subnetworks exist throughout pretraining, they are most efficiently represented earlier in pretraining. One potential explanation may be that the parameters involved in the bigram subnetwork are partially exapted to make more nuanced predictions later in pretraining, making the

---

[4]In §A.3, we discuss why we would expect bigram correlations to plateau with more allowed parameters. Of course, as subnetworks approach the full models, bigram correlations will return to the lower full model correlations (Figure 1, right).

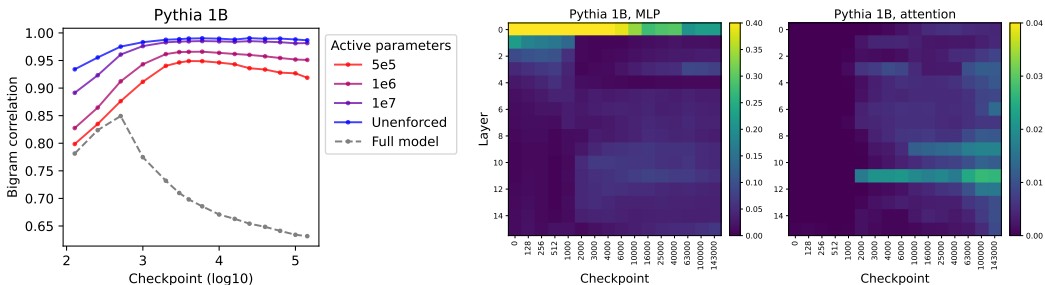

Figure 2: Left: estimated bigram surprisal correlation for bigram subnetworks with different numbers of active parameters (excluding embedding parameters) for Pythia 1B at different checkpoints (§4.1). Center, right: proportions of parameters in the Pythia 1B bigram subnetwork that are in each MLP and attention layer throughout pretraining (§4.2). Note that the color bar scale is $10\times$ larger for MLP proportions, as a far greater proportion of bigram subnetwork parameters are in the MLP layers.

bigram information less efficient to extract. Thus, the bigram distribution is optimally represented relatively early in pretraining, although still considerably *after* the full model overfits to bigrams. This optimal point might be characterized as the time when a model has fully "learned" bigrams.

## 4.2 Bigram Subnetworks Are Concentrated in the First MLP Layer

Next, we consider the distribution of active parameters in bigram subnetworks at different checkpoints. We report results for the subnetworks trained with sparsity term $\lambda = 100$ for Pythia 1B, but other models produce similar results (§A.6). As shown in Figure 2 (center, right), the plurality of bigram subnetwork parameters are in the first Transformer MLP layer, for all pretraining checkpoints. We later find that this concentration in the first Transformer layer seems to reflect an early transformation from current to next token representations in the model (§5).

Interestingly, this concentration in the first MLP layer is true even at checkpoint zero, when the model is randomly initialized. This may be because even though the model at checkpoint zero is randomly initialized, next token cross-entropy loss is used both to train the language model originally and to train all bigram subnetworks. Thus, it may be that bigram distribution learning in the first Transformer layer is an inherent property of the Transformer architecture combined with next token prediction loss, potentially due to larger gradients in earlier layers (Shleifer et al., 2021). We note that while finding bigram subnetworks in randomly initialized models might suggest that the bigram subnetworks are just arbitrary sets of parameters that happen to recreate the desired bigram predictions, the bigram correlations are lower at random initialization (Figure 2, left).[5] Furthermore, bigram subnetworks recreate important properties of the residual stream (§5), and ablating bigram subnetworks in fully trained models drastically hurts model performance (§6), suggesting that they do play an important role in language model processing.

Finally, later in training, we find that bigram subnetworks become more distributed across layers, and they have an increasing presence in attention layers (Figure 2, right; results for other models in §A.6). By the last checkpoint in Pythia 1B, 17.9% of bigram subnetwork parameters are in attention layers (82.1% in MLP). These bigram subnetwork parameters are generally spread across attention heads in each layer, but they are slightly more concentrated in attention value and output matrices (11.6% of subnetwork parameters) rather than query and key matrices (6.3% of subnetwork parameters). This is somewhat intuitive, as a bigram prediction has no reason to look back to previous tokens (the primary function of query and key matrices) because it is only dependent on the current token. The presence of bigram subnetwork parameters in attention value and output matrices might indicate that relevant bigram information is stored in those matrices (e.g. in a similar way to subject-attribute mappings that have been found in language model attention parameters; Geva et al., 2023), or it could be that these attention parameters are necessary to align the bigram subnetworks with general patterns of activation in the full language model, as we discuss in the next section.

---

[5]We find that subnetworks with high bigram surprisal correlations can be found in randomly-initialized models particularly if fully-trained token embeddings and unembeddings are patched in (§A.4).

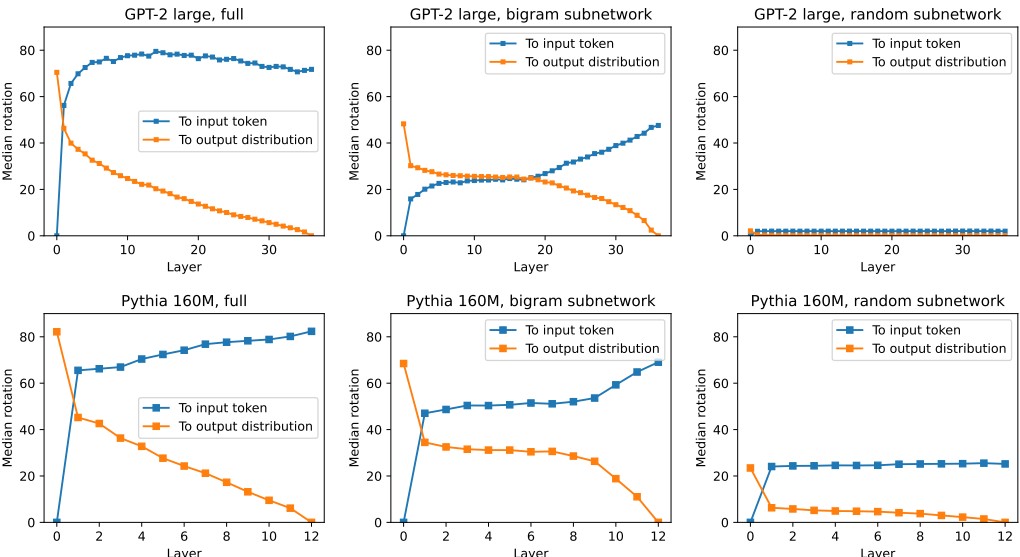

Figure 3: Median rotation to input (current token) activations and to output (next token) activations at each layer in GPT-2 large and Pythia 160M, for the full model, the bigram subnetwork, and a random subnetwork with the same size and structure as the bigram subnetwork (§5.1). In full models and their bigram subnetworks, the first layer induces a notable rotation towards next token representations.

# 5 Mapping to Next Tokens in the Residual Stream

Here, we consider how bigram subnetworks operate mechanistically in Transformer language models. Specifically, we consider the *residual stream*, the pattern of model activations across layers. Previous work has hypothesized that language models map from current to next token space in early layers (Nostalgebraist, 2020; Belrose et al., 2023). Indeed, in this section, we show that the first Transformer layer induces a sharp rotation that aligns activations with next token predictions rather than input token embeddings (§5.1). This transformation is recreated in the bigram subnetworks (§5.1), and we show that bigram subnetworks recreate many of the cross-layer similarities that characterize a language model's residual stream (§5.2).[6]

## 5.1 Rotations Induced by the First Transformer Layer

First, we consider the rotations required to map from activations at a given layer $\ell$ back to the input token activations (i.e. current tokens, after embedding) vs. to the output layer activations (next token predictions, before unembedding). Intuitively, these rotations quantify how much transformation would be required to map each layer's activations to current vs. next token representations.

For each layer, we compute activation vectors $X_\ell \in \mathbb{R}^{n \times d}$ outputted by that layer for $n =$128K tokens in context, using sequences from the OSCAR dataset as in §3. For each layer, we use ridge regression to find a linear map $L_{in} \in \mathbb{R}^{d \times d}$ that maps activations in layer $\ell$ to their original input token embeddings $X_{in}$, and a linear map $L_{out} \in \mathbb{R}^{d \times d}$ that maps activations in layer $\ell$ to their corresponding output layer activations $X_{out}$.[7] We then assess the structure of these linear transformations. A linear transformation $L \in \mathbb{R}^{d \times d}$ has $d$ complex eigenvalues including multiplicity. Because $L$ has only real entries, the eigenvalues appear in complex conjugate pairs $a \pm bi$. Each eigenvalue magnitude $||a \pm bi||$ corresponds to scaling a given input dimension, and the corresponding eigenvalue angle ($\arctan(b/a)$) corresponds to rotating that input dimension. We focus on the rotations induced by each linear transformation $L$, because scalings are intuitively less consequential for representing features in activation space; if a given feature is represented along a direction $v$, then scaling that

---

[6]Here and in later sections, we consider the bigram subnetwork trained with sparsity term $\lambda = 100$ for Pythia 1B and GPT-2 large, and $\lambda = 10$ for Pythia 160M and GPT-2 small (details in §A.3).

[7]We use ridge regression because the correlations between different activation dimensions make linear regression coefficients (i.e. entries of $L$) unstable without normalization. We use L2 regularization term $\alpha = 1$.

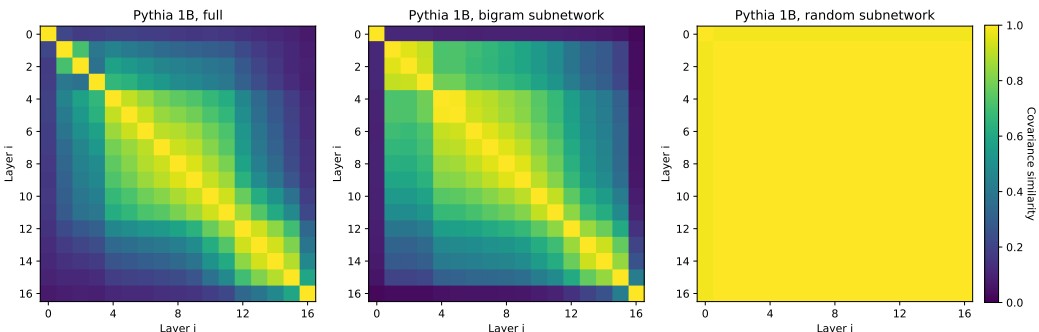

Figure 4: Cross-layer covariance similarities for Pythia 1B in the full model, its bigram subnetwork, and a random subnetwork with the same size and structure as the bigram subnetwork (§5.2). The bigram subnetwork recreates many of the patterns from the full model, despite consisting of only 0.17% of non-embedding parameters.

dimension will still encode the same information along the same direction, just multiplied by a scalar. Thus for each layer in each model, we compute the median rotation $r \in [0, 180]$ degrees for the transformation $L_{in}$ to input activations and for the transformation $L_{out}$ to output activations. Loosely, these median rotations quantify the "transformation distance" from a layer's activations to current vs. next token representations.

**Rotations from Current to Next Tokens in the First Layer.** Median rotations for $L_{in}$ and $L_{out}$ for all layers in GPT-2 large and Pythia 160M are shown in Figure 3, with results for other models in §A.7. In the full models, there is a large rotation at the first layer that aligns activations with output (next token) activations rather than input (current token) activations. For example, in GPT-2 large, the first layer increases the median rotation to the input activations from 0.0 to 56.2 degrees; it decreases the median rotation to the output activations from 70.5 to 46.3 degrees. These results align with previous work suggesting that the first layer immediately converts input token embeddings to naive next token predictions (Nostalgebraist, 2020); we quantify this effect using linear transformations fitted to many token activations.

In Figure 3 (center column), we observe that bigram subnetworks recreate this first layer transformation from current to next token space. This effect is particularly noticeable in the bigram subnetwork for Pythia 160M, where the first layer increases the median rotation to the input activations from 0.0 to 47.0 degrees; it decreases the median rotation to the output activations from 68.5 to 34.5 degrees. Importantly, a random subnetwork with the same size (and the same parameter distribution over layers and parameter blocks) does not show nearly the same degree of effect. While we observe a small first layer effect in the random subnetwork in Pythia 160M, this is likely because the Pythia bigram subnetwork is proportionally larger (8.5% of non-embedding parameters) than that of GPT-2 large (0.10% of non-embedding parameters). A larger random subnetwork is more likely to recreate some properties of the full model. Our results demonstrate that bigram subnetworks recreate the first layer transformation from current to next token space far more than would be expected from chance, even when they comprise an extremely small proportion of model parameters.

### 5.2 Covariance Similarities Between Layers

We further verify that bigram subnetworks recreate key properties of the full model's residual stream using activation covariances at each layer. Following Belrose et al. (2023), we compute pairwise similarities between covariance matrices at different layers.[8] As in Belrose et al. (2023), we drop the top singular value dimension when computing activation covariances (because this is often an outlier dimension that dominates the covariance), and we use Frobenius cosine similarity between matrices. Results for Pythia 1B are shown in Figure 4, with other models in §A.8. Not only does the bigram subnetwork recreate the discontinuity between layer zero and layer one activations (i.e. the

---

[8]While we do not expect bigram subnetwork activation covariances to mimic full model covariances themselves (because they omit many features and model operations by design), we assess whether their covariance cross-layer *similarity* matrices are similar to that of the full model.

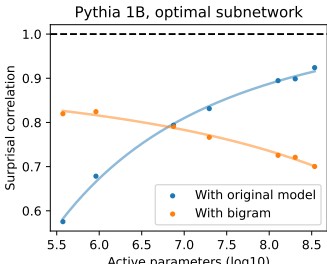
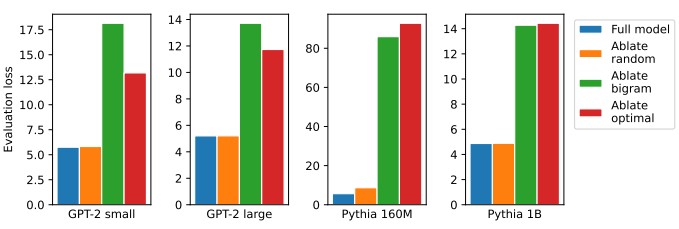

Figure 5: Left: surprisal correlations between optimal subnetworks and the original model, and between optimal subnetworks and bigram predictions, for different numbers of active parameters in Pythia 1B (§6.1). Right: language modeling evaluation loss when ablating a random subnetwork with the same size and structure as the bigram subnetwork, the bigram subnetwork itself, or an optimal subnetwork of similar size to the bigram subnetwork (§6.3).

layer zero embeddings exhibit low similarity with all other layers), the bigram subnetwork recreates patterns where certain layers block together in the full model (brighter squares in Figure 4). This is particularly notable because the Pythia 1B bigram subnetwork in Figure 4 consists of only 0.17% of model parameters. A random subnetwork of the same size (and the same distribution over parameter blocks) produces a similarity matrix with all values close to 1.0, because the subnetwork contains so few parameters that activations pass through it essentially unchanged, leading to near-perfect similarities across layers. This provides further evidence that bigram subnetworks are a minimal subset of parameters that drive core features of a language model's residual stream. We do not claim that bigram subnetworks are the *only* subset of parameters that could recreate this structure, but in the next section, we demonstrate that even if bigram subnetworks are not unique in this way, they are critical to model performance, and optimal subnetworks converge on a similar choice of parameters.

# 6   Bigram Subnetworks Approximate Optimal Subnetworks

Finally, we demonstrate through ablations that bigram subnetworks are critical to language modeling performance. We find that bigram subnetworks overlap significantly with subnetworks trained to optimally prune a model (§6.2), and ablating the bigram subnetwork hurts performance to a degree similar to ablating an "optimal" subnetwork (§6.3). This indicates that the bigram subnetworks are not just arbitrary subsets of parameters that recreate bigram probabilities by chance. Rather, they play an important functional role in making next token predictions in language models. Framed another way, bigram subnetworks are not only *sufficient* for basic next token predictions, but they are also *necessary* for reasonable next token predictions.

## 6.1   Training Optimal Subnetworks

To compare to bigram subnetworks, we train "optimal" subnetworks that seek to optimally prune a language model. We again use continuous sparsification (§3.1), but we minimize cross-entropy with the original language model output distribution rather than with the bigram distribution. As before, we train these optimal subnetworks with different sparsity terms $\lambda \in [1, 5, 10, 50, 100, 500, 1000]$. Shown in Figure 5 (left), when optimal subnetworks have more enforced sparsity (fewer active parameters), they correlate more with bigram predictions than with the original model. This indicates that bigram predictions are an efficient way to minimize next token prediction loss in constrained scenarios, even when not optimizing specifically for bigram cross-entropy. This aligns with previous work demonstrating that language models overfit to bigram predictions early in pretraining (Chang & Bergen, 2022; Choshen et al., 2022). In the next sections, when comparing bigram and optimal subnetworks, we always consider the sparsest optimal subnetwork that still contains more active parameters than the bigram subnetwork of interest.[9]

---

[9]We cannot enforce the *exact* same parameter counts as bigram subnetworks just using the sparsity term $\lambda$. As noted previously, we focus on bigram subnetworks trained with $\lambda = 100$ for Pythia 1B and GPT-2 large, and $\lambda = 10$ for Pythia 160M and GPT-2 small (details in §A.3).

## 6.2 Optimal Subnetworks Overlap Significantly with Bigram Subnetworks

Highly sparse optimal subnetworks thus approach bigram predictions in behavior. We now show that these optimal subnetworks in fact contain many of the same individual parameters as the bigram subnetworks in §3 and §4. Specifically, we compare the overlap in active parameters for optimal subnetworks and bigram subnetworks to the expected overlaps if the same number of parameters were randomly selected within each parameter block. This allows us to consider parameter overlaps after accounting for the fact that optimal and bigram subnetworks might have systematic biases to keep parameters in similar blocks.

We report results for Pythia 1B here, and we report similar results for other models in §A.9. In Pythia 1B, the bigram subnetwork and optimal subnetwork parameter overlap is $15.3\times$ greater than would be expected from chance ($p < 0.0001$; details in §A.9). In that model, 38.0% of the bigram subnetwork is contained in the optimal subnetwork, even though the optimal subnetwork is only 0.93% of model parameters and the bigram subnetwork is 0.17% of model parameters (excluding embedding parameters). This indicates that sparse optimal subnetworks are not only similar to bigram subnetworks in behavior, but they also identify similar individual model parameters.

## 6.3 Ablating Bigram Subnetworks

Finally, we evaluate language modeling loss for models when ablating the bigram subnetwork, in order to assess the *necessity* of bigram subnetworks for next token predictions. We evaluate this loss on a held out subset of 1.2M tokens from OSCAR (Abadji et al., 2021). We compare the increase in loss from ablating the bigram subnetwork to the increase in loss when ablating (1) a random subnetwork with the same size and distribution over parameter blocks as the bigram subnetwork or (2) an optimal subnetwork as described in previous sections. Shown in Figure 5 (right), ablating the bigram subnetwork results in a similar (and sometimes larger) degradation in performance to ablating an optimal subnetwork. This occurs even though we ensure that the optimal subnetworks we compare to are slightly *larger* than the corresponding bigram subnetworks (§6.1), i.e. the optimal subnetworks actually ablate slightly *more* parameters. Ablating a random subnetwork of the same size has almost no effect on language modeling performance (Figure 5, right). This indicates that bigram subnetworks are critical to language modeling performance, particularly for their small size. Sample text generations when ablating different subnetworks are reported in §A.10; ablating the bigram subnetwork leads to incoherent text generations that simply repeat a single frequent token.

## 7 Discussion and Conclusion

Our results suggest that bigram subnetworks are core components of language models. Bigram subnetworks are highly sparse, they recreate bigram predictions with $r > 0.95$, they overlap significantly with subnetworks trained to optimally prune a model, they dramatically hurt performance when ablated, and they recreate key properties of the full model's residual stream. Specifically, when added to an empty subnetwork, bigram subnetworks are *sufficient* to induce basic next token predictions (i.e. in isolation, they produce bigram predictions). Based on our ablation results, bigram subnetworks are also *necessary* for reasonable next token predictions. Finally, our enforcement of subnetwork sparsity using $\lambda$ in continuous sparsification ensures that the bigram subnetworks are *minimal* (i.e. they do not contain extraneous parameters; §A.1; §A.3; Wang et al., 2023). Together, these results provide a comprehensive view of bigram subnetworks as key language model subnetworks that are both interpretable and critical to model performance.

Speculatively, we hypothesize that bigram subnetworks emerge because early in training (or for low-performing models), bigram heuristics near-optimally decrease the language modeling loss overall, so the optimizer pushes parameters towards recreating a similar distribution. Thus, early in training (e.g. when the model overall approximates the bigram distribution), the primary function of the bigram subnetwork is likely to actually make a prediction similar to a bigram prediction. Later in training, we speculate that the model scaffolds off of these bigram predictions and subnetworks, adapting the early layers to perform generally useful early-layer computations (e.g. transforming to next token space) that still maintain some similarity to bigram predictions. It remains ambiguous whether these early-layer computations constitute "intermediate bigram predictions" (e.g. preliminary predictions before refinement by later layers; §A.5), or whether the bigram subnetwork becomes a more deeply integrated part of model processing, not separable as an "intermediate prediction".

Regardless, the persisting similarity to bigram-like computations in early layers is likely what makes it possible to extract bigram subnetworks even late in training, when these subnetworks have adapted to scaffold other computational roles as well.

Notably, even in fully trained models, the minimal and interpretable properties of bigram subnetworks make them ideal starting points from which to build language models up from component circuits. By adding specific circuits to bigram subnetworks and evaluating how model predictions change from bigram predictions, researchers can evaluate the *sufficiency* of those circuits for performing hypothesized functions. Due to the sparsity of the bigram subnetworks and the relative simplicity of the bigram function, we can be more confident that the added circuits are leveraging their own suboperations and intermediate features rather than scaffolding off of preprocessed features in an opaque language model. To facilitate these lines of research, we publicly release bigram subnetworks for GPT-2 and Pythia models up to 1B parameters. As researchers *un*-ablate interpretable circuits, we can aim to shrink the performance gap between fully interpretable model subnetworks (i.e. subnetworks built up from interpretable circuits) and their corresponding full models, setting concrete targets for mechanistic interpretability research.

## Acknowledgments and Disclosure of Funding

We would like to thank the UCSD Language and Cognition Lab for valuable discussion and feedback. Subnetworks were trained on hardware provided by the NVIDIA Corporation as part of an NVIDIA Academic Hardware Grant. Some models were also trained on the UCSD Social Sciences Research and Development Environment (SSRDE).

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

# A Appendix

## A.1 Circuit Necessity, Sufficiency, Faithfulness, Completeness, and Minimality

Although circuit *necessity* and *sufficiency* are defined briefly in §1, we define them more clearly here:

- For a circuit $C \subseteq$ model $M$, circuit *necessity* requires that ablating $C$ from $M$ removes the target behavior from $M$. Ideally, if $M$ has some more complex behavior $f_0$, and $C$ is hypothesized to perform target behavior $f_1$, then $M \backslash C$ should ideally perform $f_0 - f_1$ if there is some intuitive and well-defined way to "subtract" behavior $f_1$ from $f_0$.

- Circuit *sufficiency* requires that $C$ exhibits the target behavior when added to some minimal subnetwork $M_{min} \subseteq M$ (potentially with $M_{min} = \emptyset$). Specifically, if $M_{min}$ has some interpretable behavior $f_0$, and circuit $C$ is hypothesized to perform target behavior $f_1$, then $M_{min} \cup C$ should perform $f_0 + f_1$ if there is some intuitive and well-defined way to "add" the behaviors $f_0$ and $f_1$. This definition relates to a more general discussion of compositionality in language model circuits (e.g. Lepori et al., 2023b).

These definitions of necessity and sufficiency are closely related to the definitions of *faithfulness*, *completeness*, and *minimality* in Wang et al. (2023):

- Circuit *faithfulness* requires that $C$ replicate the target behavior as observed in the full model $M$. This is close to our definition of sufficiency, which requires that $C$ exhibit the target behavior when added to some minimal subnetwork $M_{min} \subseteq M$.

- For completeness and minimality, a parameter $v \in M$ is considered relevant to the target behavior iff there exists some $M_{ablation} \subseteq M$ such that changing $v$ affects the target behavior in $M_{ablation}$. In other words, there exists some model ablation such that $v$ affects the target behavior. We note that when there is redundancy between circuits, $v$ might affect affect target behavior only when the primary circuit is ablated, but not in the full model.

  Then, circuit *completeness* requires that all relevant parameters $v$ are included in circuit $C$. Circuit completeness implies both sufficiency and necessity. Ablating a complete circuit will remove the target behavior (necessity), because by definition, the complete circuit contains all parameters that affect the target behavior. A complete circuit alone will retain the target behavior (sufficiency), because it contains all parameters that affect the target behavior. However, completeness is difficult to evaluate in practice, because evaluating every possible subset $M_{ablation} \subseteq M$ is often intractable (Wang et al., 2023). Thus, it may be helpful to instead consider the weaker criteria of circuit sufficiency and necessity.

- Circuit *minimality* requires that $C$ contains only relevant parameters $v$ as defined above (i.e. it does not contain extraneous parameters). This is independent of both circuit sufficiency and circuit necessity. For example, the full model $M$ itself is technically both necessary and sufficient for all observed behaviors in $M$. However, the full model $M$ is not minimal for many target behaviors. This is because while $M$ overall is necessary for each target behavior, not all of its component subcircuits are necessary for each target behavior. An ideal circuit would be necessary, sufficient, *and* minimal for a target behavior.

Using these criteria, bigram subnetworks are *necessary* for next token predictions (ablating them drastically hurts language modeling performance; §6.3), *sufficient* for bigram predictions (in isolation, they produce bigram predictions; §3.2), *faithful* to bigram predictions (bigram surprisal correlations $r > 0.95$; §3.2), and *minimal* with respect to bigram predictions (enforced sparsity $\lambda$ in continuous sparsification optimizes the subnetwork to drop irrelevant parameters; §A.3). However, the bigram subnetworks are not necessarily *complete*. There may be other redundant parameters in the model that could also recreate bigram predictions. Luckily, to use bigram subnetworks as minimal subnetworks onto which to add other circuits (e.g. to evaluate the sufficiency of other circuits for other behaviors; §7), we do not need the bigram subnetworks to be complete as in Wang et al. (2023). We simply need a single (potentially not unique) subnetwork that produces interpretable and minimal next token predictions and that recreates key internal structures from the full model.

## A.2 Subnetwork Training Details

As described in §3.1, we use continuous sparsification (Savarese et al., 2020; Lepori et al., 2023b) to identify subsets of model parameters that minimize cross-entropy loss with the bigram distribution.

We estimate the bigram distribution by counting bigram frequencies in 1.28B tokens of OSCAR web text (10M sequences of 128 tokens; Abadji et al., 2021), tokenized with the corresponding model tokenizer. Because the corpus is large enough that all tokens appear at least once (i.e. all possible bigram "prefixes"), there are no undefined bigram probabilities in our experiments, and we do not need to smooth the bigram distribution estimate.

To train the subnetworks themselves, we build upon the implementation in Lepori et al. (2023a). For all fully trained models (§3) and when training "optimal" subnetworks (§6), we train subnetworks with sparsity terms $\lambda \in [0, 1, 5, 10, 50, 100, 500, 1000]$ (Equation 1). For checkpoints during pretraining (§4), we train subnetworks with $\lambda \in [0, 1, 10, 100, 500]$. Subnetworks are trained on sequences of 128 tokens with batch size 32 and learning rate 5e-5, with a multiplicative sigmoid temperature decrease of 1.001 per training step (dictating how fast the continuous mask approaches a binary mask). We learn the subnetwork mask over all model parameters except the token embedding, token unembedding, and layernorm parameters.[10] All mask parameters are initialized at $0.0$ before the sigmoid ($0.50$ after the sigmoid).

For each subnetwork, we stop training when the number of mask parameters in $[0.10, 0.90]$ (i.e. "undecided" mask parameters) is less than 1% of the number of mask parameters greater than $0.90$ (i.e. parameters to be included in the subnetwork). This ensures that the number of "undecided" parameters will not change the learned subnetwork by more than 1% when the continuous mask is converted to a binary mask. In practice, it takes roughly 5K to 10K training steps to reach this convergence criterion. To verify training stability, we check for any spikes in cross-entropy loss of $+0.25$ or $\times 1.25$. If such a spike is not recovered at least 100 steps before the end of training, we re-train the subnetwork. Each subnetwork takes approximately four to twelve hours to train on a single NVIDIA RTX A6000 (48GB) GPU, depending primarily on model size. Subnetwork training costs make up the vast majority of computational costs to produce the results in this paper.

### A.3 Bigram Subnetwork Selection

In §4.2, §5, and §6, when analyzing the bigram subnetwork for a specific model, we focus on the bigram subnetwork with maximal sparsity but before any notable drop in bigram surprisal correlation. Our analyses in those sections include GPT-2 small, GPT-2 small re-trained (checkpoints from Chang et al., 2024), GPT-2 large, Pythia 160M, and Pythia 1B. For each fully trained model, of the bigram subnetworks trained with different sparsity terms $\lambda$, we select the sparsest subnetwork (highest $\lambda$) that is still within $0.04$ correlation of the subnetwork trained with $\lambda = 0$ (no sparsity enforced).

This aims to ensure that the subnetworks we analyze only contain parameters that actually contribute to the subnetwork's behavior, even if these are before the subnetwork has fully plateaued in bigram surprisal correlations (around 10M active parameters; Figure 1). With more allowed parameters (lower $\lambda$), the subnetwork has more flexibility to optimize the bigram cross-entropy loss (Equation 1), but it is also more likely to include extraneous parameters that have little effect on the subnetwork's behavior regardless of whether they are included or not. For example, in a sparse subnetwork, it would not matter whether certain parameters $\theta_A$ are included if they only write some feature to a subspace that is read by some parameters $\theta_B$ that are already ablated (Merullo et al., 2024b). Indeed, when $\lambda = 0$ (no sparsity enforced), we find that most subnetworks consist of roughly 50% of model parameters, indicating that most parameters do not have a significant pressure to be included or not included in the subnetwork.

Thus in our analyses in §4.2, §5, and §6, for GPT-2 (re-trained), GPT-2 small, and Pythia 160M, we use the bigram subnetwork trained with $\lambda = 10$. For GPT-2 large and Pythia 1B, we use the bigram subnetwork trained with $\lambda = 100$. The parameter counts and bigram surprisal correlations for these subnetworks are reported in Table 1. As noted in §6, for optimal subnetworks, we consider the sparsest optimal subnetwork that still contains more active parameters than the bigram subnetwork selected for that model. In general, we find that our results remain qualitatively similar for different $\lambda$, as long as the bigram subnetworks are reasonably sparse (e.g. $< 10\%$ of parameters active).

---

[10]We do not include embedding and unembedding (language model head) parameters in the mask because we hypothesize that embedding parameters for rare tokens might be highly likely to get masked. Specifically, when optimizing for bigram cross-entropy loss and encouraging sparsity, the continuous sparsification method might be likely to find it optimal to simply drop many embedding and unembedding parameters for rarer tokens, because they contribute less to the loss.

| Model | Sparsity $\lambda$ | Active parameters | Total mask parameters | Bigram correlation $r$ |
|---|---|---|---|---|
| Pythia 160M | 10 | 7238986 | 84934656 | 0.953 |
| Pythia 1B | 100 | 1378087 | 805306368 | 0.959 |
| GPT-2 small* | 10 | 924639 | 84934656 | 0.962 |
| GPT-2 small | 10 | 904866 | 84934656 | 0.952 |
| GPT-2 large | 100 | 727556 | 707788800 | 0.946 |

Table 1: Subnetwork parameter counts for the bigram subnetworks that we focus on for our analyses in §4.2, §5, and §6. GPT-2 small* indicates the GPT-2 small replication from Chang et al. (2024). Details in §A.3.

| Model | Full model | Bigram subnetwork | Subnetwork, at random initialization | Subnetwork, random except embeddings | Embeddings, empty subnetwork | Embeddings, linear transform |
|---|---|---|---|---|---|---|
| Pythia 160M | 0.690 | **0.964** | 0.787 | **0.965** | 0.305 | 0.615 |
| Pythia 1B | 0.632 | **0.987** | 0.896 | **0.986** | 0.628 | **0.936** |
| GPT-2 small* | 0.680 | **0.987** | 0.759 | **0.974** | 0.680 | **0.921** |

Table 2: For different models, we report bigram surprisal correlations for (1) the full model, (2) the bigram subnetwork, (3) the bigram subnetwork found in a randomly initialized model, (4) the bigram subnetwork found in a randomly initialized model with trained token embeddings and unembeddings patched in, (5) the embeddings and unembeddings alone with an empty subnetwork, and (6) the embeddings and unembeddings with a learned linear transformation between them. Subnetworks here use $\lambda = 0$ (no sparsity enforced), to maximize bigram surprisal correlations. Correlations above $r = 0.90$ are bolded. GPT-2 small* indicates the GPT-2 small replication from Chang et al. (2024). Details in §A.4.

## A.4 Importance of Trained Token Embeddings

Interestingly, we find that bigram subnetworks with high bigram surprisal correlations can be found in randomly-initialized models, as long as fully-trained token embeddings and unembeddings are patched in (Table 2; "subnetwork, random except embeddings", c.f. "subnetwork, at random initialization"). For example, we can find a subnetwork with bigram surprisal correlation $r = 0.986$ in Pythia 1B when all parameters are randomly initialized except the token embedding and unembedding parameters. This suggests that trained token embeddings are much of the reason we can find bigram subnetworks in fully trained models, but not in randomly initialized models.

However, keeping only the token embedding and unembedding parameters (and layernorm parameters) does not result in high bigram surprisal correlations (Table 2; "embeddings, empty subnetwork"). In other words, the embeddings do not recreate the $W_U W_E$ embedding-unembedding bigram structure described for zero-layer Transformers in Elhage et al. (2021). This is not entirely surprising, but it indicates that the bigram subnetwork parameters we find in Transformer layers are still necessary for high bigram correlations. For example, these non-embedding subnetwork parameters might be responsible for triggering the bigram information encoded in the embedding and unembedding parameters, and thus these non-embedding parameters are still important components of the bigram subnetwork.

The fact that much information is contained in token embeddings and unembeddings may also be relevant for interpreting *tuned lens* results in other work. The tuned lens learns a linear transformation between layer $\ell$ activations and the unembedding matrix, to elicit intermediate model predictions from layer $\ell$ (Belrose et al., 2023). Equivalently, all remaining Transformer layers are replaced with a learned linear transformation. However, we find that such a linear transformation can often recreate bigram predictions even directly from the input embeddings (Table 2; "embeddings, linear transform"; bigram surprisal correlation $r > 0.90$ for Pythia 1B and GPT-2 small), suggesting that a linear transformation can sometimes extract desired predictions regardless of whether they are truly "intermediate predictions" of the model (§A.5).[11] This would make the tuned lens an overly

---

[11] For the linear transformation from input embeddings to bigram distribution activations (before unembedding), we use the transformation $L_{out}$ from §5.1 for layer zero activations in the bigram subnetwork. This transformation

strong probe with potential false positive "intermediate predictions", because the learned linear transformation can significantly rotate and rescale the input embeddings to match a target output distribution. We note from Figure 3 in the main text that even if a linear transformation can recreate the output distribution from layer zero, it requires a significant amount of rotation of the activations to do so.

## A.5 Do Language Models Make Intermediate Bigram Predictions?

While bigram subnetwork predictions are highly correlated with bigram predictions when other parameters are set to zero, this does not necessarily mean that fully trained language models "make bigram predictions" that are hidden within the full model. Defining whether a model makes an "intermediate prediction" at some layer $\ell$ has often been cast as early-exiting from the model, where the token unembedding matrix is applied directly to activations after layer $\ell$. Unfortunately, this untuned *logit lens* is relatively unreliable in early layers (Belrose et al., 2023), where bigram subnetworks are concentrated. In contrast, a *tuned lens* (Belrose et al., 2023) that learns a linear transformation between layer $\ell$ activations and the unembedding matrix might be too strong a probe: above, we find that a tuned lens can reach bigram surprisal correlations of $r > 0.90$ even when applied directly to input token embeddings, because the learned linear transformation is able to significantly rotate and rescale the input embeddings to match a bigram output distribution (§A.4). In other words, an untuned logit lens is too *weak* a probe to find many intermediate predictions (many false negatives), but a tuned logit lens may be too *strong* a probe (many false positives); a clear definition of an "intermediate prediction" after layer $\ell$ remains nebulous.[12] In our work, we do not claim that language models make intermediate bigram predictions in early layers, but we note that our results in §5 (demonstrating a transformation towards next token space in the first layer) would align with this hypothesis. Determining where, if anywhere, intermediate next token predictions are encoded in language model activations is an interesting direction for further research.

---

has been fitted to recreate the subnetwork output activations (before unembedding) from the layer zero input embeddings; because the subnetwork is trained to recreate bigram predictions, the subnetwork output activations are close to the bigram distribution. Thus, $L_{out}$ approximately maps from input embeddings to bigram distribution outputs before unembedding.

[12]In a similar way, one might consider continuous sparsification to be too strong a probe for finding bigram subnetworks; however, our ablation and optimal pruning results (§6), along with mechanistic analyses (§5), suggest that the bigram subnetworks we find are not just arbitrary subsets of parameters.

### A.6 Checkpoint Results for Other Models

Results as in Figure 2 but for GPT-2 small (re-trained, checkpoints from Chang et al., 2024) and Pythia 160M are shown in Figure 6. As in Pythia 1B in the main text (§4), the bigram subnetworks peak in bigram surprisal correlations over one thousand steps after the full model begins to diverge from bigram predictions, but the bigram subnetworks become less efficiently compressed later in pretraining. Throughout pretraining, the plurality of subnetwork parameters are in the first MLP layer, but the subnetworks spread more to other layers (including attention layers) later in pretraining.

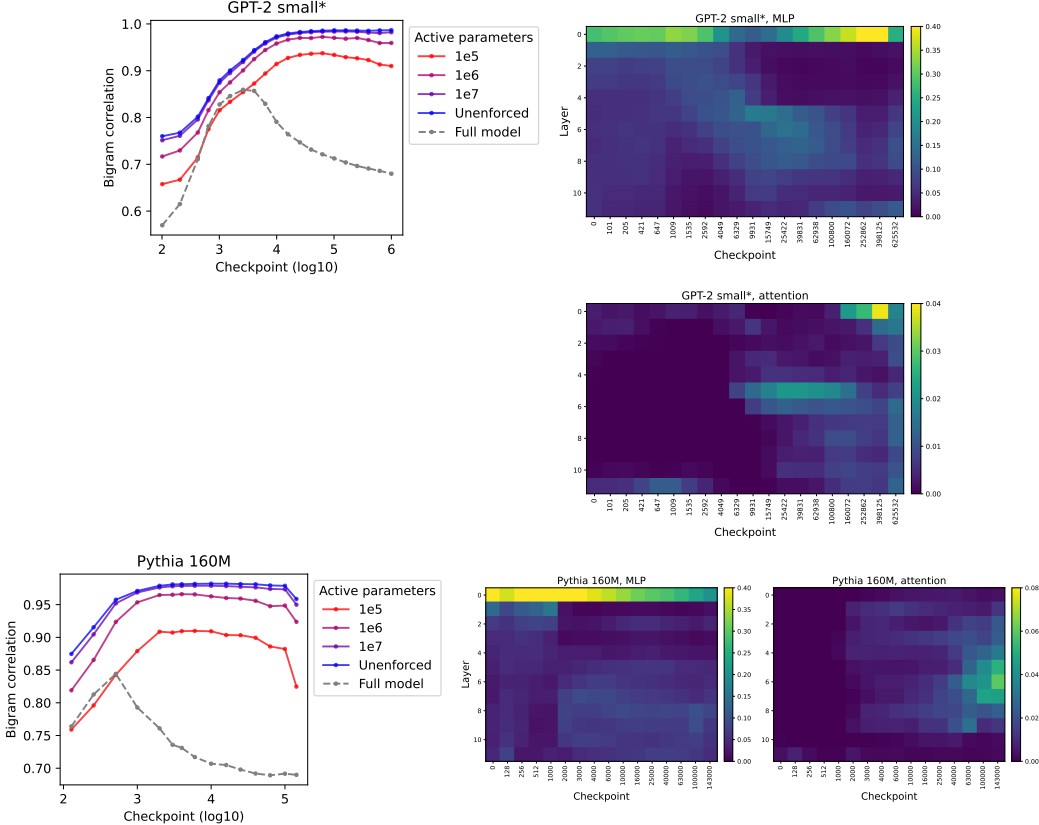

Figure 6: Results as in Figure 2, but for GPT-2 small* and Pythia 160M. GPT-2 small* indicates the GPT-2 small replication from Chang et al. (2024) with checkpoints. Left: estimated bigram surprisal correlation for bigram subnetworks with different numbers of active parameters (excluding embedding parameters) at different checkpoints. Right: proportions of parameters in the subnetwork that are in each MLP and attention layer throughout pretraining. Note that the color bar scale is 5× or 10× larger for MLP proportions, as a far greater proportion of bigram subnetwork parameters are in the MLP layers.

### A.7 Input and Output Rotation Results for Other Models

Results as in Figure 3 but for other models are shown in Figure 7. As in GPT-2 large and Pythia 160M in the main text (§5.1), in the full models, the first layer induces a large rotation that aligns activations with output (next token) activations rather than input (current token) activations. This effect is recreated in the bigram subnetworks, although to a smaller degree in Pythia 1B. However, in all cases, the bigram subnetwork recreates the pattern from the full model to a much larger degree than a random subnetwork (with the same size and the same distribution over parameter blocks as the bigram subnetwork) does.

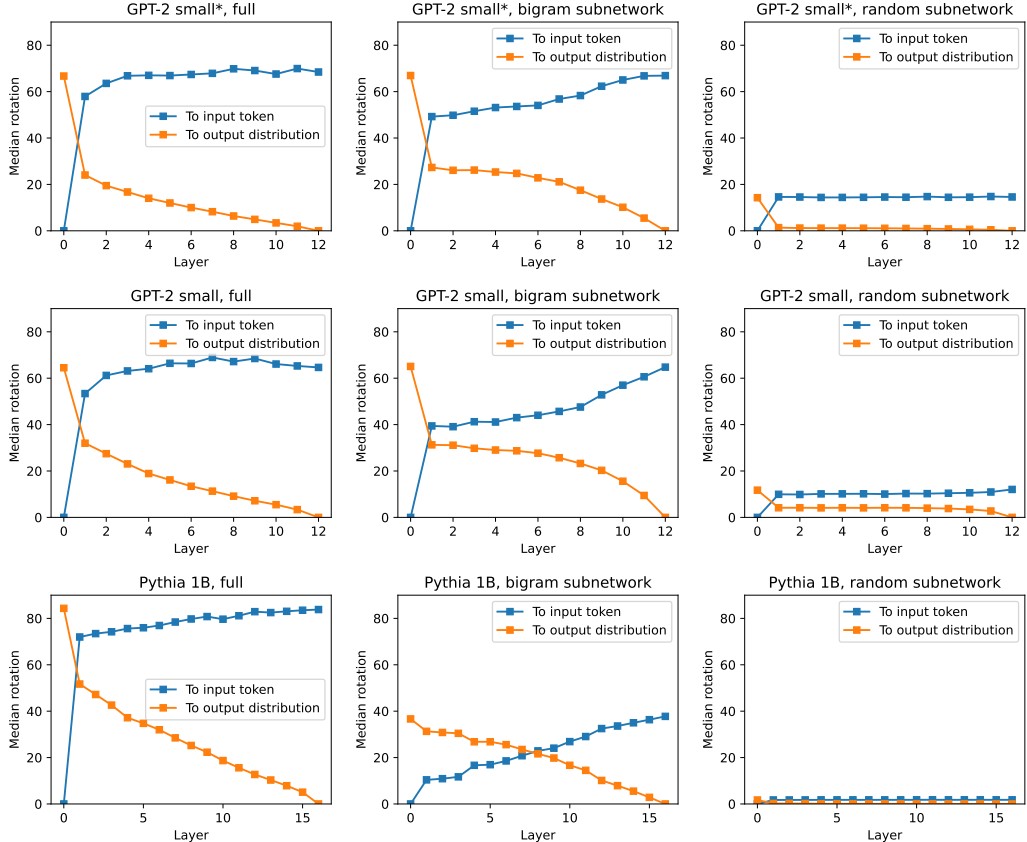

Figure 7: Results as in Figure 3, but for other models. These show the median rotation to input (current token) activations and to output (next token) activations at each layer, for the full model, the bigram subnetwork, and a random subnetwork with the same size and structure as the bigram subnetwork. GPT-2 small* indicates the GPT-2 small replication from Chang et al. (2024).

## A.8 Covariance Similarity Results for Other Models

Results as in Figure 4 but for other models are shown in Figure 8. As in Pythia 1B in the main text (§5.2), the bigram subnetworks recreate many cross-layer covariance similarity patterns from their corresponding full models, despite consisting of only a small proportion of model parameters. Random subnetworks with the same size (and the same distributions over parameter blocks) as the bigram subnetworks generally do not recreate these patterns, producing similarity matrices with values almost entirely near 1.0 (consistent with results in the main text; §5.2). We do observe that the discontinuity between layer zero and layer one is somewhat recreated in the random subnetworks for smaller models. This is likely because the bigram subnetworks are a larger proportion of model parameters in the smaller models; the corresponding random subnetworks (which are matched for bigram subnetwork size) then contain more parameters and are thus more likely to recreate patterns from the full models.

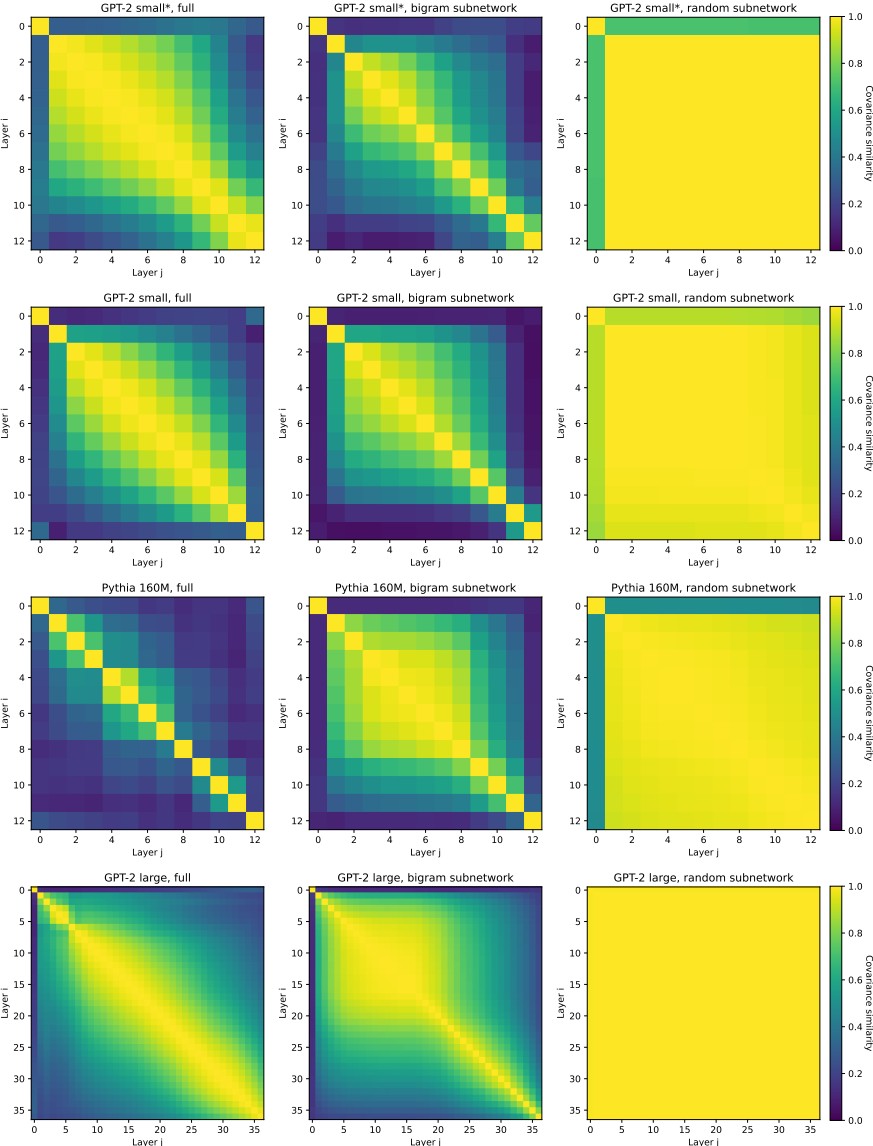

Figure 8: Results as in Figure 4, but for other models. These show cross-layer covariance similarities for the full model, its bigram subnetwork, and a random subnetwork with the same size and structure as the bigram subnetwork. The bigram subnetworks recreate many patterns from the full models, much moreso than random subnetworks of the same size. GPT-2 small* indicates the GPT-2 small replication from Chang et al. (2024).

## A.9 Optimal Subnetwork Results for Other Models

Results as in Figure 5 (left) but for other models are shown in Figure 9. As in Pythia 1B in the main text (§6.1), when more sparsity is enforced (fewer active parameters), optimal subnetwork surprisals correlate more with bigram surprisals than with the original model surprisals. We do observe a slight drop in bigram correlations for the sparsest optimal subnetworks in Pythia 160M; this is somewhat expected, because in extremely sparse scenarios, bigram correlations drop even for the bigram subnetworks themselves (Figure 1, left, center). Optimal subnetwork correlations with bigram predictions still remain considerably higher than correlations with the full model in these sparse scenarios.

In Table 3, we report the actual and expected (by chance) parameter overlaps between the bigram subnetwork and the optimal subnetwork for each model, as described in §6.2 (with bigram subnetworks selected as in §A.3). Concretely, when considering the parameter overlap between two subnetworks $C_0$ and $C_1$, we generate 10K random subnetwork pairs (pairs of parameter masks) that randomly select the same number of parameters as $C_0$ and $C_1$ respectively within each parameter block. Then, we count the number of random pairs whose parameter overlap is greater than or equal to the parameter overlap of $C_0$ and $C_1$. This gives us the probability that two random subnetworks would have equal or greater overlap than $C_0$ and $C_1$, after accounting for the number of parameters kept per block in $C_0$ and $C_1$. As in Pythia 1B in the main text (§6.2), for all models, the actual overlaps between bigram subnetworks and optimal subnetworks are far larger than would be expected by chance. Specifically, in all cases, none of the 10K randomly generated pairs have greater parameter overlap than the bigram subnetwork and the optimal subnetwork (and thus $p < 0.0001$).

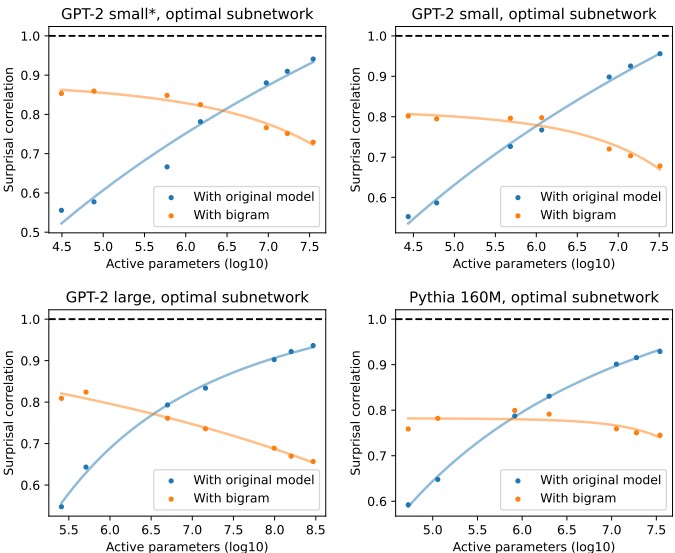

Figure 9: Results as in Figure 5 (left), but for other models. These show surprisal correlations between optimal subnetworks and the original model, and between optimal subnetworks and bigram predictions, for different numbers of active parameters. GPT-2 small* indicates the GPT-2 small replication from Chang et al. (2024).

| Model | Bigram subnetwork | Optimal subnetwork | Overlap | Expected overlap | Actual / expected | % bigram contained |
|---|---|---|---|---|---|---|
| Pythia 160M | 8.52e-2 | 1.33e-1 | 5.23e-2 | 1.37e-2 | 3.81× | 61.4% |
| Pythia 1B | 1.71e-3 | 9.30e-3 | 6.51e-4 | 4.26e-5 | 15.27× | 38.0% |
| GPT-2 small* | 1.09e-2 | 1.77e-2 | 2.41e-3 | 3.43e-4 | 7.03× | 22.1% |
| GPT-2 small | 1.07e-2 | 1.37e-2 | 1.53e-3 | 2.10e-4 | 7.27× | 14.3% |
| GPT-2 large | 1.03e-3 | 7.06e-3 | 3.34e-4 | 2.63e-5 | 12.70× | 32.5% |

Table 3: Proportions of total model parameters (excluding embedding parameters) in (1) the bigram subnetwork (selected as in §A.3), (2) an optimal subnetwork of roughly similar size (§6.1), (3) the overlap between the two, and (4) the expected overlap between the two by chance, then (5) the multiplier by which the actual overlap exceeds the expected overlap, and (6) the percentage of the bigram subnetwork that is contained in the optimal subnetwork. GPT-2 small* indicates the GPT-2 small replication from Chang et al. (2024).

## A.10 Text Generations After Subnetwork Ablations

In Table 4, we report sample text generations for Pythia 1B and GPT-2 large for (1) the original model, (2) the bigram subnetwork, (3) ablating the bigram subnetwork, and (4) ablating a random subnetwork with the same size (and the same distribution over parameter blocks) as the bigram subnetwork. All text generations here use sampling temperature $\tau = 0.30$. In line with the evaluation loss results in §6.3, ablating the bigram subnetwork drastically hurts text generation performance. Ablating a random subnetwork of the same size and structure has little, if any, noticeable effect. We observe qualitatively similar results for the smaller models.

| Model | Subnetwork | Text |
|---|---|---|
| Pythia 1B | Full model | This is a sentence that I have been told by my friends and family members.  It is a sentence that I have heard from my teachers. |
| | Bigram subnetwork | This is a sentence that the most of the most popular in the first time.  The first time to be a few days, and the best way to see the best way to the same time to the first time. |
| | Bigram ablated | This is a sentence that ....  ..  .... ...............  ...  .............  .....  .. ......  ....................  ....... |
| | Random ablated | This is a sentence that I have written in my own words, and I have tried to express my thoughts in a way that is understandable to you.  I will not be able to get over the fact that I have been a victim of the system. |
| GPT-2 large | Full model | This is a sentence that is not true.  The only thing that is true is that the world is not a perfect place. There are things that are not perfect. |
| | Bigram subnetwork | This is a sentence that the best way to be the best to the same time.  The most of the first time.  The first round and the best way to the best possible. |
| | Bigram ablated | This is a sentence that \n \n \n \n \n \n \n \n \n \n \n \n \n \n \n \n \n \n \n \n \n \n \n \n \n \n \n \n \n \n \n \n \n \n |
| | Random ablated | This is a sentence that is often used in the context of the "right to be forgotten" in the EU. "The right to be forgotten" is a legal right that allows you to request that a website remove or obscure your personal information from search results. |

Table 4: Sample text generations for Pythia 1B and GPT-2 large for the full model, the bigram subnetwork, ablating the bigram subnetwork, and ablating a random subnetwork with the same size and structure as the bigram subnetwork (§A.10). Input prompt text is in purple. All generations use sampling temperature $\tau = 0.30$.

