# OpenReview forum: "Bigram Subnetworks: Mapping to Next Tokens in Transformer Language Models"
_NeurIPS.cc/2025/Conference — NeurIPS 2025 spotlight_

### Official Review · Reviewer_FaeD · 2025-06-19

**Clarity:** 4
**Significance:** 3
**Originality:** 3
**Rating:** 5
**Confidence:** 3

**Summary:**

The paper focuses on a basic heuristic that is expected to be common in language modelling, bigram modelling. It finds a circuit in models that represent it, and studies the traits of those circuits.

**Questions:**

na

**Ethical Concerns:**

["NO or VERY MINOR ethics concerns only"]

**Final Justification:**

Was good before is good still... Not sure why this mandatory field is needed, or why neurips requires so much work that affects the quality of the communicated work so little.

**Quality:**

4

**Strengths And Weaknesses:**

Strengths
The motivation, contributions, method and findings are clear and clean, clean enough that even non-experts would be able to follow.
The finding about the consistent size needed to represent this knowledge is extremely interesting.
The findings open thought on many future avenues, from what does it mean that after learning little is decreased, to the consistent size, to reinitialization.
Weaknesses:
The found parameters are mainly on the first layer which reduces the strength of the claims, it makes it less clear wthat the model uses it at the beginning when it overfits bigrams (with all parameters) and such issues.

Minor:
l26 wouldn't it be better for the citations to be near the topic they refer to, instead of all together?
In Fig. 1 why not write e^x instead of just x? Isn't it  an easier way of conveying the number and that it is log scale
l127 it is shown in the left or center not on the right figure.

---

> ### Author Rebuttal · Authors · 2025-07-31
>
> Thank you for the helpful comments!
>
> > The found parameters are mainly on the first layer which reduces the strength of the claims, it makes it less clear wthat the model uses it at the beginning when it overfits bigrams (with all parameters) and such issues.
>
> Yes, we find that bigram subnetwork parameters are consistently concentrated in the first layer. Early in training when the model overfits to bigrams, it may be that the rest of the model is still essentially untrained, and the noise of the later layer parameters washes out, leaving the early layer bigram predictions unchanged even if the later layer parameters are included. This result is closely related to model distillation and capacity measurement, as it indicates that for un-fully-trained models, a substantial proportion of parameters can be dropped without changing model behavior.
>
> > Minor: l26 wouldn't it be better for the citations to be near the topic they refer to, instead of all together?
>
> Good point, we will update this!
>
> > In Fig. 1 why not write e^x instead of just x? Isn't it an easier way of conveying the number and that it is log scale.
>
> We used x instead of 10^x to avoid having the exponents too small to read, but we can update this in the revised manuscript.
>
> > l127 it is shown in the left or center not on the right figure.
>
> Raw surprisal correlations between bigram subnetworks and the bigram distribution are reported in the table on the right of Figure 1. We agree that the left and center also show these effects, so we will update this mention in the revised manuscript.

---

> > ### Comment · Reviewer_FaeD · 2025-08-01
> >
> > Don't feel obliged to make the paper worse if the feedback wasn't helpful (e.g. if you don't manage to make the figure better looking), it is not about my feelings it is about helping you communicate your work better or (in this work less so) make better work.

---

### Official Review · Reviewer_qApJ · 2025-06-28

**Clarity:** 2
**Significance:** 2
**Originality:** 3
**Rating:** 5
**Confidence:** 3

**Summary:**

This work investigates a subnetwork that emulates bigram language model behavior from larger neural language models, as a starting point to explore more complex circuits. The experiments with continuous sparsification yield intriguing findings, for example, (i) the existence of bigram subnetwork, (ii) the substantial role of the first MLP layer, and (iii) the significant overlap with optimal subnetworks.

**Questions:**

Quesitons
- How did you estimate the gold bigram distribution in Eq.1?
    - Is there any concern that the findings are biased toward the bigram estimation method? (e.g., reporting bias, smoothing...)
    - Are you targeting bigram at the subword level or word level?
- Which parameters are the target of masking in Eq. 1? Does it also include parameters in the language model head?
    - I’m curious about whether the first layer parameters indeed have bigram knowledge or if it just triggers some bigram knowledge in the language model head.
- What type of statistical test yielded p<0.0001 in Section 6.2?

**Ethical Concerns:**

["NO or VERY MINOR ethics concerns only"]

**Final Justification:**

I agree with other reviewers to accept this paper and clarify my stance toward acceptance. My main concerns were on the impact and clarification of methodological details, which are minor points to improve the quality of this paper and not subject to the reason for rejection.

**Limitations:**

yes

**Quality:**

3

**Strengths And Weaknesses:**

Strength:
- The perspective to bigin with presumably simplest circuit of bigram prediction is unique and interesting.
- The analysis is extensive in the sense that the subnetwork is inspected from multiple perspectives, including training dynamics, layer-wise analysis, and geometric interpreation of internal activations.

Weakness:
- Technical details are not elaborated carefully (see Questions)
- Existing facts, such as logit-lens can extract somewhat reasonable probabilities even from the first layer (ll.234—237), and the gradual contextualization from earlier to later layers [1] may already imply the findings —the first layer is responsible for weakly contextualized, bigram prediction— of this paper. In this sense, this paper would not be so informative to the community
- Language model prediction head (and its interplay with activations) should also be an important analysis target, given the fact that it handles primitive language statistics [2]. However, it’s not explicitly analyzed or discussed. (see Questions)

[1] Brunner, Gino, et al. "On Identifiability in Transformers." International Conference on Learning Representations.
[2] Kobayashi et al., Transformer Language Models Handle Word Frequency in Prediction Head. Findings 2023.

---

> ### Author Rebuttal · Authors · 2025-07-31
>
> Thank you for pointing out these areas to clarify!
>
> > Which parameters are the target of masking in Eq. 1? Does it also include parameters in the language model head? I’m curious about whether the first layer parameters indeed have bigram knowledge or if it just triggers some bigram knowledge in the language model head... Language model prediction head (and its interplay with activations) should also be an important analysis target, given the fact that it handles primitive language statistics. However, it’s not explicitly analyzed or discussed.
>
> We learn the subnetwork mask over all model parameters except the token embedding, token unembedding (language model head), and layernorm parameters (Section 3.1, Appendix A.2). We agree that the language model head contains important information about primitive language statistics (e.g. token frequencies and bigram probabilities). Indeed, in Appendix A.4, we find bigram subnetworks in randomly-initialized models as long as trained token embeddings and unembeddings are patched in. This indicates that, as the reviewer suggests, the first layer parameters might primarily trigger some bigram knowledge in the embedding and unembedding parameters; still, our ablation results indicate that this triggering of bigram knowledge is a key part of Transformer language model processing, and thus these non-embedding subnetwork parameters are still a central component of the bigram subnetwork.
>
> We also note that we did not include embedding and unembedding (language model head) parameters in the mask because we hypothesized that embedding parameters for rare tokens might be highly likely to get masked. Specifically, when optimizing for bigram cross-entropy loss and encouraging sparsity, the continuous sparsification method might be likely to find it optimal to simply drop many embedding and unembedding parameters for rarer tokens, because they contribute less to the loss.
>
> > How did you estimate the gold bigram distribution in Eq.1? Are you targeting bigram at the subword level or word level?
>
> Thank you for pointing out that we omitted this detail! Bigram distributions were estimated at the subword (token) level for each model, using 10M sequences of 128 tokens (1.28B tokens total) from the OSCAR English corpus of web text. Estimates were computed by counting bigram frequencies in the corpus. Because the corpus was large enough that all tokens appeared at least once (i.e. all possible bigram "prefixes"), there were no undefined bigram probabilities in our experiments. Thus, we did not smooth the bigram distribution estimate. We will include these details in the updated manuscript.
>
> > Is there any concern that the findings are biased toward the bigram estimation method? (e.g., reporting bias, smoothing...)
>
> Reported above, we did not use any smoothing for the bigram distribution estimate. However, we agree that there may be reporting bias: we estimated the bigram distributions from OSCAR English web text, but the models were not all trained on exactly this dataset. Still, both the Pythia and GPT-2 models are primarily trained on English web text, so we expect the bigram distributions in their training datasets to be roughly similar to our estimates from OSCAR.
>
> > What type of statistical test yielded p<0.0001 in Section 6.2?
>
> When considering the parameter overlap between two subnetworks A and B, we generate 10,000 random subnetwork pairs (pairs of parameter masks) that randomly select the same number of parameters as A and B respectively within each parameter block. Then, we count the number of random pairs whose parameter overlap is greater than or equal to the parameter overlap of A and B. This gives us the probability that two random subnetworks would have equal or greater overlap than A and B, after accounting for the number of parameters kept per block in A and B. In all cases, none of the 10,000 randomly generated pairs had greater parameter overlap than A and B, i.e. $p < 0.0001$. We will include these details in the updated manuscript.

---

> > ### Comment · Reviewer_qApJ · 2025-08-04
> >
> > I appreciate the responses from the authors. These details should be elaborated or more highlighted in the main part of the paper, even though some are written in Appendix. In particular, I re-read Appendix A.4, and find that the discussion there is fruitful in handling my concern. To make my stance clearer, I'd raise my score toward acceptance.

---

### Official Review · Reviewer_AhmT · 2025-06-30

**Clarity:** 4
**Significance:** 3
**Originality:** 3
**Rating:** 5
**Confidence:** 4

**Summary:**

The authors consider the existence and ubiquity of "bigram subnetworks", which are subcomponents of a large language model which are used to predict the next token given ONLY the previous one (hence "bi-gram"). They constructively demonstrate the existence of such networks in large language models, show that they are tiny yet critical for model performance, and perform mechanistic analysis on the behavior of such subnetworks. Indeed, they show that such networks are (mostly, approximately) what you get when you try to aggressively prune a LLM.

**Questions:**

- What practical implications may exist as a result of identifying bigram subnetworks, that don't exist when identifying the optimal subnetworks (using the definitions in the paper)?

**Ethical Concerns:**

["NO or VERY MINOR ethics concerns only"]

**Final Justification:**

I choose to keep a score of 5/6 (Accept). All reviewers (including me) are generally satisfied and impressed by the paper, particularly the thorough analysis and solid methodology. The rebuttal did not change this in either direction. The one remaining large improvement to the paper I can imagine is some demonstration of practical applications; the rebuttal hints at some ways to do this, but the authors do not actually implement one (or commit to implementing one), which is an issue common to this style of work. Overall, the paper is a solid work and strong contribution to the field of interpretability, so I recommend acceptance.

**Limitations:**

yes

**Paper Formatting Concerns:**

No concerns

**Quality:**

4

**Strengths And Weaknesses:**

Strengths:
- The experimental design is in general well done. The experiment suite clearly isolates the existence, dynamics, and importance of the bigram subnetworks.
- The findings are interesting and serve as an exemplar of scaling fine-grained analysis of subnetworks of large models.
- The study is very detailed (especially the mechanistic components).
- The paper is clearly written and easy to read.

Weaknesses:
- The work may have some implications for practice (say for pruning), and it would be helpful and make the paper more impactful to include explicit discussion of them. This would give the work some practical relevance on top of being a nice scientific work.

---

> ### Author Rebuttal · Authors · 2025-07-31
>
> Thank you for the helpful comments!
>
> > The work may have some implications for practice (say for pruning), and it would be helpful and make the paper more impactful to include explicit discussion of them. This would give the work some practical relevance on top of being a nice scientific work... What practical implications may exist as a result of identifying bigram subnetworks, that don't exist when identifying the optimal subnetworks (using the definitions in the paper)?
>
> We hope that the bigram subnetworks would be useful as starting points for interpretable circuits that trigger behaviors that diverge from bigram predictions (Section 7). The bigram subnetworks allow researchers to disentangle the subnetwork for basic next token prediction (the bigram subnetwork) from an actual behavior of interest. In particular, unlike an optimal subnetwork, the bigram subnetwork's predictions are highly interpretable, so we can be more confident that the bigram subnetworks are not performing opaque operations that might confound results for other circuits. Finding these more precise and interpretable circuits could facilitate model pruning methods that better target specific behaviors rather than "optimal" behavior as a whole.

---

> > ### Comment · Reviewer_AhmT · 2025-08-02
> > **Reply to Rebuttal**
> >
> > Thanks for clarifying a potential application. My understanding is that incorporating this approach (decoupling bigram networks from more interesting circuits) into current interpretability pipelines is non-trivial. One way to make this work really stand out may be to empirically demonstrate that this approach really can lead to better interpretability results. As is, the work is already quite strong and so I keep my score (recommending acceptance).

---

### Official Review · Reviewer_2aPg · 2025-07-01

**Clarity:** 4
**Significance:** 4
**Originality:** 4
**Rating:** 5
**Confidence:** 2

**Summary:**

This paper presents an analysis of "bigram subnetworks" in several pretrained LLMs, subsets of ~10M parameters discovered by continuous sparsification which generate predictions similar to those of an optimal bigram model. These parameters are concentrated in early LM layers, and despite forming a small fraction of overall model parameters, replicate geometric / statistical properties of the full model's residual stream. The authors argue play a necessary role in language model behavior as well: they overlap significantly with subnetworks discovered by general model pruning, and ablating bigram subnetworks causes major degradation in quantitative and qualitative measures of sample quality.

**Questions:**

See above.

**Ethical Concerns:**

["NO or VERY MINOR ethics concerns only"]

**Final Justification:**

The reviewers seem to agree that this is a conceptually and technically interesting paper worth accepting. My minor questions were adequately addressed by the authors in their response. I will keep my score as is, 5/6.

**Limitations:**

Yes

**Quality:**

4

**Strengths And Weaknesses:**

This is an exemplary mechanistic interpretability paper, beginning with a clear theoretical proposal that is supported by two arms of experiments (demonstrations of sufficiency and necessity). I agree with the authors' argument that this kind of result establishes a clear baseline from which we can work to develop more complex descriptions of language model behavior that depart from *n*-gram-like computations. I also appreciate the conceptual work in the appendix, both in the definitions of circuit properties and in the discussion of what it would mean for a model to actually make "intermediate predictions."

I find little to criticize in this paper. The only slightly surprising set of results is those around random initialization: L174 (bigram subnetworks are concentrated in the first layer even when we have random weights) and §A.4 L608 (we can discover bigram subnetworks which perform equally well in randomly initialized models, as long as we patch embedding/unembedding matrices).
- L608 makes some sense to me — we do expect embeddings/unembeddings themselves to encode basic bigram statistical properties, and subsampling the axes of a high-dimensional random projection could preserve that.
- L174 suggests (if I understand correctly?) that the concentration of subnetwork parameters in early layers is not necessarily due to an actual computation localized in these layers, and may be an artifact of the architecture+objective+optimization setting.
- Together, these findings cut down the (mostly implicit?) intuition that these bigram subnetworks are performing a more meaningful computation, e.g. as discussed in §A.5.

I would appreciate if the authors could add at least a bit of summary speculation late in the paper about what these findings mean for possible computational functions of the "bigram subnetwork."

---

> ### Author Rebuttal · Authors · 2025-07-30
>
> Thank you for the thorough review!
>
> > L174 suggests (if I understand correctly?) that the concentration of subnetwork parameters in early layers is not necessarily due to an actual computation localized in these layers, and may be an artifact of the architecture+objective+optimization setting.
>
> Yes, we agree that the concentration of bigram subnetwork parameters in early layers in the randomly-initialized networks suggests that such a property might be an artifact of the architecture+objective+optimization setting. Other subsets of parameters in other layers might recreate similar properties, and we do not necessarily claim that the bigram subnetworks we find are unique (l. 265-268; 552-557), despite their apparent importance to language model predictions.
>
> Still, we speculate that as models compress more information into their parameters throughout training, the bias towards early layers for bigram computations might make the models likely to compress any redundant bigram-like computations into those early layers. This could compound with the fact that later model computations could scaffold off of bigram computations (e.g. refining the bigram predictions), making it more optimal for bigram computations to appear in earlier layers. Future work might investigate whether these types of suboperation dependencies and parameter location biases explain the tendencies for particular suboperations to appear in specific layers (e.g. to better understand the learning mechanisms driving classic results such as "BERT Rediscovers the Classical NLP Pipeline"; Tenney et al., 2019).
>
> > Together, these findings cut down the (mostly implicit?) intuition that these bigram subnetworks are performing a more meaningful computation, e.g. as discussed in Appendix A.5. I would appreciate if the authors could add at least a bit of summary speculation late in the paper about what these findings mean for possible computational functions of the "bigram subnetwork."
>
> We speculate that early in training (or for low-performing models), bigram heuristics near-optimally decrease the language modeling loss overall, so the optimizer pushes parameters towards recreating a similar distribution. Thus, early in training (e.g. when the model overall approximates the bigram distribution), the bigram subnetworks are likely indeed serving a computational function similar to bigram prediction. Later in training, we speculate that the model scaffolds off of these bigram predictions and subnetworks, adapting the early layers to perform generally useful early-layer computations (e.g. transforming to next token space) that still maintain some similarity to bigram predictions (even if not exactly an "intermediate bigram prediction", as discussed in Appendix A.5). It may be that this persisting similarity to bigram predictions in early layers is what makes it possible to extract bigram subnetworks even late in training, when these subnetworks have adapted to scaffold other computational roles as well.

---

### Decision · Program_Chairs · 2025-09-17

**Decision:**

Accept (spotlight)

**Comment:**

The paper investigates minimal mechanisms in transformer LLMs for predicting the next token from the current one ("bigram subnetworks"). Through several careful experiments, they study necessity and sufficiency of these subnetworks in basic next-token prediction, and identify the first layer MLPs as the key component responsible for these prediction tasks. The implications of these findings for more complex mechanistic analyses remain to be explored, but overall the reviewers were all very happy with the paper and the discussion, thus I recommend acceptance.